# Effectiveness of Social Measures against COVID-19 Outbreaks in Selected Japanese Regions Analyzed by System Dynamic Modeling

**DOI:** 10.3390/ijerph17176238

**Published:** 2020-08-27

**Authors:** Makoto Niwa, Yasushi Hara, Shintaro Sengoku, Kota Kodama

**Affiliations:** 1Graduate School of Technology Management, Ritsumeikan University, Osaka 567-8570, Japan; mak.niwa@po.nippon-shinyaku.co.jp; 2Discovery Research Laboratories, Nippon Shinyaku Co., Ltd., Kyoto 601-8550, Japan; 3TDB Center for Advanced Empirical Research on Enterprise and Economy, Faculty of Economics, Hitotsubashi University, Tokyo 186-8603, Japan; yasushi.hara@r.hit-u.ac.jp; 4Life Style by Design Research Unit, Institute for Future Initiatives, the University of Tokyo, Tokyo 113-0033, Japan; sengoku.s.aa@m.titech.ac.jp; 5Center for Research and Education on Drug Discovery, The Graduate School of Pharmaceutical Sciences in Hokkaido University, Sapporo 060-0812, Japan

**Keywords:** COVID-19, system dynamics, new infectious disease

## Abstract

In Japan’s response to the coronavirus disease 2019 (COVID-19), virus testing was limited to symptomatic patients due to limited capacity, resulting in uncertainty regarding the spread of infection and the appropriateness of countermeasures. System dynamic modelling, comprised of stock flow and infection modelling, was used to describe regional population dynamics and estimate assumed region-specific transmission rates. The estimated regional transmission rates were then mapped against actual patient data throughout the course of the interventions. This modelling, together with simulation studies, demonstrated the effectiveness of inbound traveler quarantine and resident self-isolation policies and practices. A causal loop approach was taken to link societal factors to infection control measures. This causal loop modelling suggested that the only effective measure against COVID-19 transmission in the Japanese context was intervention in the early stages of the outbreak by national and regional governments, and no social self-strengthening dynamics were demonstrated. These findings may contribute to an understanding of how social resilience to future infectious disease threats can be developed.

## 1. Introduction

### 1.1. COVID-19 Pandemic and Reactions by Society

Coronavirus disease 2019 (COVID-19), a respiratory disease caused by a novel coronavirus that initially emerged in the city of Wuhan at the end of 2019 [1], has quickly spread all over the world. Because of its novelty, which means the lack of specific medicine for it, the dominant countermeasures are isolation and supportive medication. This indicates that a large number of hospital beds will be needed. Thus, control strategies such as early diagnosis, isolation, and hospitalization are essential. A lack of strategy can lead to the collapse of the healthcare system if it is overwhelmed by patients.

In handling the complex COVID-19 transmission processes in the population and the effects of societal factors, the idea to use system dynamics, describing complex social systems as a collective set of mathematical equations, was drawn based on some considerations. First, a stock-flow model in system dynamics adequately describes population transition, including delay in time course. Moreover, the effects of social factors on disease transmission can be mathematically modeled with minimal complexity. Second, a causal loop diagram, also used in system dynamics, can describe feedback systems in society, which is important in social reaction.

System dynamics have often been used in health systems, and in addressing health problems such as obesity, diabetes, hypertension, mental health, mortality, smoking, infectious diseases, injury by violence, respiratory diseases, substance abuse, disability, quality of life, and maternal and child birth complications [2]. As an infectious disease, human immunodeficiency virus (HIV) transmission has been studied. Batchelder et al. conceptualized the effects of social and ecological conditions affecting women at risk of HIV [3]. Weeks et al. studied the effects of the HIV test and treatment care continuum on community viral load [4]. None of these studies addressed the quantitative features of disease transmission. On the other hand, a subpopulation or sector frame-based quantitative model has been used as a subcomponent of the model, such as population change over time in an obesity study [5] or demographics of the elderly in a health care systems study [6]. Based on these, the use of a population-based stock-flow model to describe disease transmission was thought to be feasible.

This study primarily aimed to clarify the effect of intervention by a modeling approach. In addition, the study sought to explore social factors of the effectiveness of the control of new infectious diseases. As an overall approach, a COVID-19 epidemic case in Japan in spring 2020 was analyzed using system dynamic modeling.

### 1.2. Previous Studies and Research Questions

This study examines four research questions. (1) Is a modeling approach effective in overcoming the lack of information (actual number of infected patients)? (2) What are the most important measures to prevent the spread of infection? (3) What are the factors that influence infection among societal factors (in demography and behavior)? (4) What is important in the future response to new infectious diseases?

There are several studies dealing with measures against COVID-19. Yan et al. summarized various countermeasures by different authorities [1]. Summarized recommendations are mainly on behavior (washing hands, keeping rooms ventilated and sanitized, wearing masks, avoiding social activities, staying away from crowded areas, and observing social distancing). Dickens et al. used an agent-based model to test the effectiveness of home-based and institutional isolation. The analysis clarified the usefulness of institutional containment and risks of home-based isolation [7]. Gerli et al. investigated the lockdown effort of European countries, and pointed out the importance of timeliness of lockdown [8].

Still, no studies encountered the ambiguity of the COVID-19 situation in Japan. Making the best use of the flexibility and simplicity of system dynamics, this study aims to grasp the whole picture of the COVID-19 outbreak in Japan using abundant information on demography and behavior. To detect potential regional differences, three regions that have enough confirmed cases and have urban cities were analyzed. They were Tokyo (the capital), the Osaka prefecture, and the Hokkaido prefecture. To avoid complexity, prefectures that have satellite cities were not analyzed.

### 1.3. Analytical Approach Applied in This Study

As analytical approaches, the following five analyses were performed. First, the effects of medical countermeasures were illustrated using causal loop analysis. Second, a stock flow model describing the mass of infected population was developed to analyze the dynamics of infection. Third, the effectiveness of actions preventing the saturation of medical capacity was tested by simulation. Fourth, the relationships between transmission reduction efficiency and regional differences in social factors were explored. Finally, the effects of social factors on disease preventive behavior were analyzed using causal loop analysis to provide suggestions for a sustainable society beyond new infectious diseases.

## 2. Materials and Methods

### 2.1. Data on COVID-19 Epidemic in Japan

The number of confirmed positives by polymerase chain reaction (PCR) virus testing by day was obtained from local governments [9,10,11] and summarized.

### 2.2. Data on Societal Factors

Population, airport arrivals from foreign countries, and the number of employees working in the region were obtained from Japanese Government Statistics [12]. Reduction in outings was derived from a published analysis by the National Institute of Informatics, which utilized the location information of mobile phones [13]. The number of companies by industries and the average number of employees by industries located in the region of interest were provided by Teikoku Databank (TDB), available through the TDB Center for Advanced Empirical Research on Enterprise and Economy (TDB-CAREE), Hitotsubashi University, Tokyo, Japan. TDB is a major corporate credit research company in Japan that collects various corporate data through door-to-door surveys. Around 1700 field researchers visit and interview firms to obtain corporate information in every industrial category and location.

### 2.3. Modeling and Software

Causal loop and stock-flow models were built using Vensim PLE (Ventana Systems Inc., Harvard, MA, USA). Data summary was performed using Microsoft Excel (Office 365; Microsoft Corporation, Redmond, WA, USA).

### 2.4. Causal Loop Model

To visualize the interrelations between variables, causal loop diagrams were constructed. Components recognized by preceding studies or prior knowledge were included as variables, and they were logically interrelated.

As medical components, infection, contact, inapparent infection, preventive behavior, recovery, susceptible proportion [14], coming from another region, limitation of diagnosis, the challenge of mass screening, lack of hospital capacity, inadequate medication, and deaths were included.

As social components, intervention by national or local government [3], awareness raising regarding physical distancing and hygiene measures [15], countermeasures by private or public enterprises [16], new business practices, working from home, and contact reduction in commuting were included.

### 2.5. Stock Flow Model

Stock flow models regarding population as a stock were built to describe the epidemic dynamics in three eminent epidemic regions: the Tokyo metropolitan area, Osaka prefecture, and Hokkaido prefecture. The timeframe was set after 11 March 2020, in order to capture the COVID-19 outbreak in late March and April 2020. As a stock (population), susceptible, newly infected, inapparently infected (mild symptoms), having moderate symptoms or developing diseases, having serious symptoms, having non-serious symptoms, isolated, hospitalized, recovered, and dead population were prepared.

Inbound virus carriers (in incubation period) were assumed to have arrived before 3 April 2020, when virus testing for all airport arrivals was started. The number of inbound carriers was estimated as 746,525 in March 2020 [12] and the assumed positive rate 0.003. The estimated carriers were then divided in proportion to the population of the region. The flow between stocks was parameterized according to the scenario depicted in Figure 1. Flow rate was mathematically expressed as the quotient of the sub-population divided by duration of flow. Parameters given exogenously are shown in Table 1. Only patients with serious symptoms were assigned to virus testing, as was done in Japan, and the theoretical ratio of all infected patients to virus tested patients was set to 5 based on the given parameters. The effective reproduction number was left as an endogenous variable and estimated from manual curve fitting against the number of diagnosed patients over the time (representative data is shown in Appendix A). The model was qualified by observing the consistency of dimension in the process of model building and visual inspection of agreement between predicted and real positive patient numbers.

## 3. Results

### 3.1. Causal Loop Analysis of Medical Systems

The causal loop diagram is shown in Figure 2. There were two clinically important endpoints: infection and deaths. Most causal loops for infection were self-intensifying loops; thus, reducing the intensity was important. Connection points were contacts; thus, reducing such contacts was important. The only antagonistic effect loop was the reduction of susceptible rate via increase in recovered population. A strategy to intensify this antagonistic loop (i.e., obtaining of social immunity) was partially possible, but an increase in recovery was supported by an increase in infection rate. An increase in infection rate intensified disease transmission through increased contact and at the same time caused inadequate isolation of non-diagnosed patients and inadequate medication, meaning the possibility of overflow of medical systems. Regarding death, the causal loops were simple. Less infection, improved hospital capacity, better medication, and improved virus testing efficiency were important for reducing deaths.

### 3.2. Quantitative Analysis by Stock-Flow Analysis

#### 3.2.1. Prediction of Transmission without Intervention

The initial (baseline) isolation effect was parameterized by manual curve fitting to the actual diagnosed patient number in the early phase. In Tokyo, the isolation effect was negligible at baseline. This indicates that transmission efficiency was almost the same as expected from the basic reproduction number. In Osaka and Hokkaido, a two-phase spread was assumed from the increased patient number. The isolation effect was reduced 10% in Osaka and 68% in Hokkaido.

After parameterization, transmission without intervention was simulated. In Tokyo, infection spread out rapidly and overwhelmed most of the population in 150 days (Figure 3). Virus testing and hospitalization was completely saturated.

Comparison of the actual patient number and the simulation is shown in Figure 4. Clear divergence was observed, and the effect of intervention is suggested. Based on the scale in the X-axis of the graph, the state of emergency was declared on day 27, and was expanded to cover the whole nation on day 36.

#### 3.2.2. Effects of Interventions on Transmission Efficiency

To simplify the analysis, the effects of interventions were combined with isolation effect and the parameter; transmission efficiency was computed as an endogenous parameter. Curve fitting was done in a sequential manner, from earlier timeframe to later timeframe, assuming that intervention was done sequentially. After the curve fitting in the intervention phase, transmission efficiency under intervention was reduced by 75% (25% remaining efficiency) in Tokyo, which started in early April (when the state of emergency was declared) (Figure 5, left). In Osaka, in addition to the first reduction in early April, the second phase started at the end of April when the Japanese holiday season began. Transmission efficiency was reduced by 60% and 85% (40% and 15% remaining efficiency) in the first and second phase, respectively (Figure 5, center). In Hokkaido, where a regional state of emergency had been declared in February 2020 and was lifted in the middle of March, transmission efficiency decreased to 68% (32% remaining efficiency) at the end of March. Subsequently, it increased to 90% at the end of April (10% remaining efficiency), two weeks after the inclusion of the region in the state of emergency (in the middle of April) and at the beginning of the holiday season (Figure 5, right). Net remaining transmission efficiency relative to baseline under the best intervention was 25%, 17%, and 31% in Tokyo, Osaka, and Hokkaido, respectively.

Under this real condition, no overflow in virus testing or hospital capacity was observed.

### 3.3. Simulation Using Stock-Flow Model

#### 3.3.1. Quarantine at Airports

Starting from 3 April 2020, all airport arrivals were tested for the virus and isolated if they were virus positive. This ideally meant that no more inbound virus carriers were joining the community. The effect of this quarantine was investigated by simulation. To simplify the analysis, no false negatives were assumed.

Simulation was run on an assumption that airport arrivals in March 2020 continued and that the positive rate stayed flat at 0.003. As a result, a 1%, 2%, or 8% increase in the number of infected patients at the end of April was expected. In Tokyo and Osaka, where countermeasures for transmission were effective, no serious impact of inbound virus carriers was expected by simulation. In Hokkaido, where controls in April were not effective, inbound carriers moderately affected the community.

#### 3.3.2. Effect of Timeliness of the State of Emergency

A previous study indicated that a delay in locking down cities results in the rapid spread of the disease [8]. Thus, the effect of delayed intervention was simulated. A comparison was made on all infected patients simulated, as confirmed positives do not reflect the real infected population in the current situation.

Simulations were run on Tokyo and Osaka, which were the targets of the first phase of the state of emergency. Delaying the intervention for one week resulted in 140% and 75% more total infections in Tokyo and Osaka, respectively (the Osaka case was simulated with a holiday effect). This result shows a similar but slightly lesser impact of intervention delay reported in a previous study; generally, reporting an 11-day delay results in 10 times the mortality.

Under this scenario, an overflow in virus testing and hospital capacity was observed.

### 3.4. Effects of Societal Factors

To examine the potential effects of societal factors on COVID-19 transmission, the relationship between several societal factors and transmission was explored. Factors potentially related to disease transmission are shown in Table 2.

As a semi-quantitative observation, baseline transmission efficiency seemed to be related to population density, as supported by the theory described in [21]. The maximum intervention effect was similar across the regions. The intervention effect before the holiday season seemed related to a reduction in outings.

Although Tokyo has the most workers and most companies’ headquarters, outings were efficiently reduced before the holiday season. This good response in Tokyo can be partly attributed to companies in the capital placing emphasis on business continuity [22], thus facilitating working from home. In contrast, the reaction in Hokkaido after the expansion of the state of emergency was relatively slow, and the transmission efficiency declined two weeks after the expansion.

### 3.5. Causal Loop Diagram for Societal Factors

The interrelationship of societal factors and disease preventing behavior is shown in Figure 6. The potential relationship suggested in the previous section was considered. Some factors are hypothetical. In this case, a noticeable effect of preventing behavior was observed, and intervention by government was apparent. This implies that intervention by government was essential and no self-strengthening dynamics were noticeable in the society in the early phase of the outbreak.

## 4. Discussion

The rapid spread of COVID-19 is threatening health systems with capacity challenges. The United States, with the largest number of patients as of March 2020, seems to be challenged by a healthcare capacity problem [23]. The inpatient bed occupancy rate varies by region, with the highest being 79% (in Maryland, May 2020) [24]. The highest intensive care unit (ICU) bed occupancy rate is 84% (in the District of Columbia, May 2020).

Japan, with only 7.3 beds per 100,000 inhabitants [25], is one of the countries that suffer from hospital bed shortage [26]. As of April 2020, there were 12,500 beds for novel infectious diseases nationwide, while there were 10,000 patients in Japan [27]. Although 31,383 hospital beds were ensured by 21 May 2020 [28], health systems are still at risk. Some local governments even have plans to provide care to low-risk patients in hotels. Under this condition, a control strategy to reduce the peak number of patients remains important.

To reduce the transmission as an effort to delay and lower the epidemic peak, governments imposed restrictions on movement in local communities. Many countries, such as China, Italy, the US, and the UK, locked down their cities to prevent the spread of the disease. In Japan, a state of emergency was declared on 7 April 2020, giving authorities the power to enforce stay-at-home orders and to close businesses. Although Japanese authorities did not describe the countermeasures as a lockdown, prefectural authorities asked people to refrain from traveling across prefectures, unnecessarily going out, and to stay away from public gatherings [29]. In addition, all schools were closed. Initially, this affected the capital, Tokyo, and six other prefectures (Saitama, Chiba, Kanagawa, Osaka, Hyogo, and Fukuoka). Subsequently, it was expanded nationwide on 16 April 2020.

Information on the effectiveness of these interventions is warranted, but the whole context of what is happening is not well understood because of the limited testing capacity. No one knows the actual number of patients infected. To overcome this, the use of a structured model with an apparent/inapparent infection ratio and efficiency of virus testing was considered.

This study primarily analyzed COVID-19 transmission dynamics and the effects of initial measures by governments. A special feature of the current model is that it includes symptom rate, virus testing capacity, and hospital capacity, which were major concerns in the early phase of the COVID-19 outbreak. The current model heavily depended on demographic data and has fewer accompanying variables in comparison to prior system dynamics studies [4]. Consideration of more detailed variables, especially health-protective behaviors such as the practice of hygiene or physical distancing measures, may help identify important factors in basic societal systems regarding disease prevention. In addition, the association of human activity and temperature or humidity is possible [30]. Specific research on each component to provide detailed information is warranted.

In the causal loop diagram, the importance of reducing contacts was highlighted. A strategy to obtain social immunity by allowing infection is theoretically possible; however, realistically, allowing infection leads to an increase in deaths through an increase in disease transmission and inadequate medication. Stock and flow analysis confirmed that an increase in infections overwhelms healthcare systems.

The stock-flow model adequately described the dynamics of the COVID-19 outbreak in three Japanese regions. Baseline isolation effect in the early phase was negligible in Tokyo, little in Osaka, and considerable in Hokkaido. Primarily, this could be related to population density as supported by transmission theory [21] and observations in the United States [30]. The hypothetical mechanism for the associations between population density and transmission proposed by Rubin et al. is increased droplet transmission and potentially airborne transmission in close proximity [30]. After the state of emergency declaration in April, transmission efficiency of the disease markedly decreased to 17–31% of the baseline. In this case, the disease was primarily controlled by national and local government interventions. The most important measure was the reduction of contacts in the early phase of the outbreak by national and local governments.

Attempts to build a causal loop diagram for interrelationship analysis revealed that no self-strengthening dynamics were noticeable in the society. This indicates that interventions by the government were essential in the meantime. As a potential reinforcing loop, a loop with new business practice and awareness raising regarding physical distancing and hygiene measures was hypothesized. Any other well-recognized component did not construct any reinforcing loops. Further investigation to confirm the self-strengthening dynamics, beginning with new business practice, and efforts to strengthen such dynamics are warranted for a sustainable society.

The strength of this study is the use of SD techniques. Stock-flow modeling is relatively simple, but it was effective in showing the overall dynamics of virus transmission when virus testing was inadequate. Stock-flow modeling also enabled estimation of the impact of interventions. Further simulation is possible for virus testing efficiency, hospital capacity, and a new medicine.

The limitations of this study are as follows. The stock-flow model utilized simple arithmetic operations and described the average dynamics of a population. This does not adequately describe the probability process that should be demonstrated by more complex models or multi-agent models. The model was constructed based on the fundamental monitor and control strategy in Japan but detailed approaches may have slightly changed over time based on local government’s policy. In addition, model validity was not fully investigated although basic validity, such as dimension consistency and consistency of predicted and real positives, was checked. The approach of leaving one parameter as endogenous made extensive validation somewhat challenging. This could be overcome by comparing multiple regions as external validation; however, no other region in Japan has enough patients to be used in building models with comparative accuracy. Nevertheless, the model sufficiently described the outbreak of COVID-19 in three Japanese regions and was useful in describing the early phase of the outbreak. A more precise investigation should be conducted in the future for the development of science. The basic structure of the current stock-flow model reflects Japanese national response strategy for COVID-19 to limit virus testing to patients with obvious symptoms for better use of diagnostic resources. This makes comparing the effectiveness of measures across countries difficult, which is an important theme with this new disease. Nevertheless, this approach enabled determination of the possible effects caused by saturation of virus testing, which was important in analyzing the effectiveness of measures undertaken by Japanese authorities. Further analysis using newly collected epidemic data and more detailed social activity data is warranted in the future.

## 5. Conclusions

This study primarily highlighted the importance of reducing contacts via causal loop and stock-flow model analysis. Moreover, the importance of interventions by government in the early phase of new infectious diseases was emphasized, as no reinforcing loop to act against infection was found in the society. Exploration of self-strengthening dynamics, beginning with new business practice and efforts, is warranted for a sustainable society.

## Figures and Tables

**Figure 1 ijerph-17-06238-f001:**
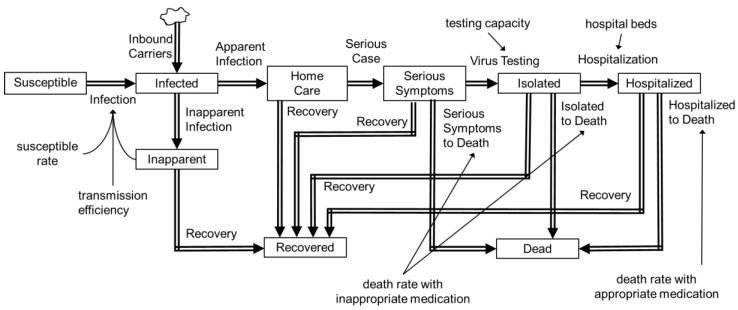
Stock-flow model for coronavirus disease 2019 (COVID-19) transmission.

**Figure 2 ijerph-17-06238-f002:**
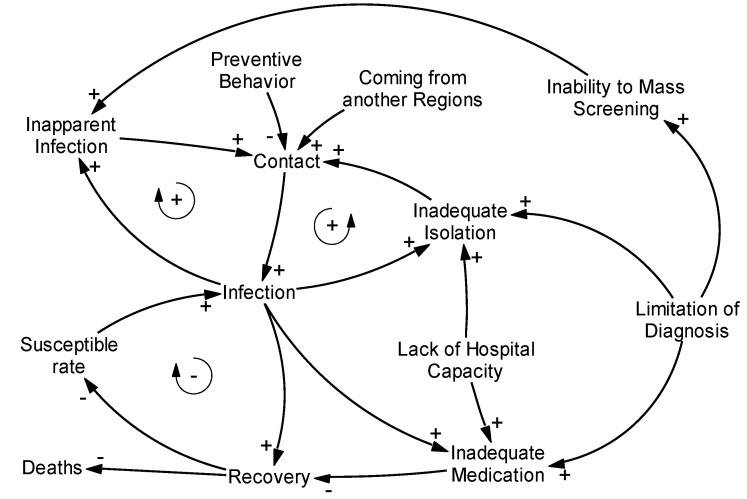
Causal loop diagram of the COVID-19 outbreak.

**Figure 3 ijerph-17-06238-f003:**
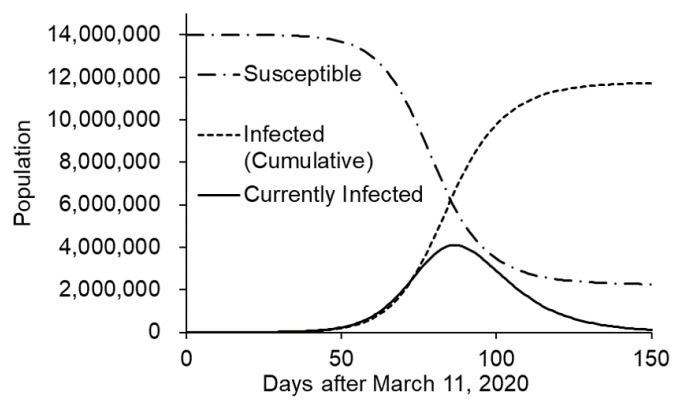
Simulation of the COVID-19 outbreak without intervention (Tokyo case).

**Figure 4 ijerph-17-06238-f004:**
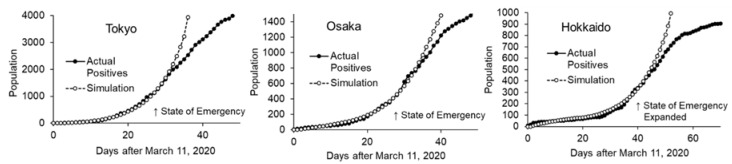
Simulation without intervention and actual patient numbers in 3 regions.

**Figure 5 ijerph-17-06238-f005:**
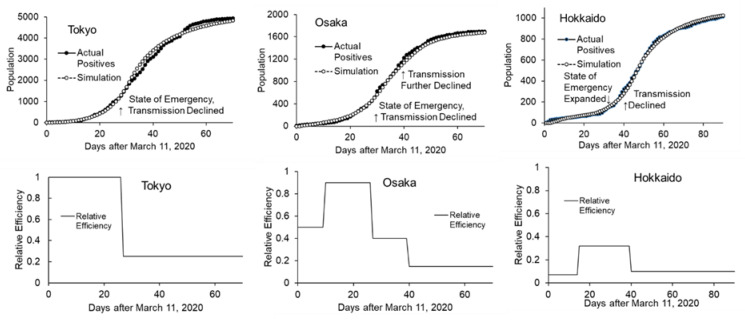
Actual and simulated confirmed positives in 3 regions (**upper**) and estimated transmission efficiencies (expressed as relative to efficiency derived from natural reproduction rate, **lower**) are shown. The periods with maximum transmission efficiency in each region were considered as the baseline of outbreak.

**Figure 6 ijerph-17-06238-f006:**
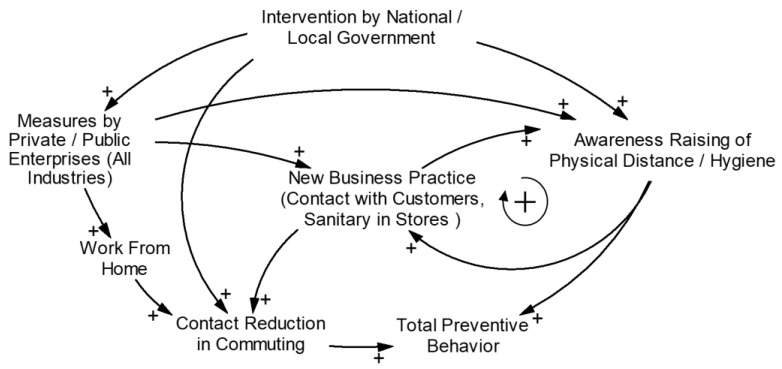
Causal loop diagram for societal factors related to new infectious diseases.

**Table 1 ijerph-17-06238-t001:** Exogenously given parameters in 2020 spring outbreak scenario.

Parameters	Value	Basis
Susceptible People	Start with 14,000,000 (Tokyo), 8,800,000 (Osaka), and 5,300,000 (Hokkaido)	Whole population, 1 October 2019 [12]
Incubation period	5 days	From the literature [17]
Inbound virus carrier	8 (Tokyo), 5 (Osaka), and 3 (Hokkaido) daily before 3 April 2020	All foreign arrivals multiplied by assumed positive rate (0.003) were divided in proportion to the population
Baseline positives per day before epidemic	5 (Tokyo), 7 (Osaka), and 7 (Hokkaido)	Assumed from average from 11 March 2020 to 15 March 2020
Initial value, incubation period, related to inbound	24 (Tokyo), 15 (Osaka), and 9 (Hokkaido)	Inbound carrier assuming 3 days of incubation period remaining
Initial value, incubation period, domestic	125 (Tokyo), 175 (Osaka), and 175 (Hokkaido)	Baseline positives divided by serious symptoms rate (0.2) and multiplied by incubation days (5)
Initial value, mild symptoms	140 (Tokyo), 175 (Osaka), and 175 (Hokkaido)	Baseline positives divided by serious symptoms rate (0.2), multiplied by recovery days (14) and inapparent rate (0.4)
Initial value, moderate or developing symptoms	30 (Tokyo), 42 (Osaka), and 42 (Hokkaido)	Baseline positives divided by serious symptoms rate (0.2), multiplied by development days (2) and apparent rate (0.6)
Initial value, serious symptoms	5 (Tokyo), 7 (Osaka), and 7 (Hokkaido)	Baseline positives multiplied by assumed diagnosis day 1
Days to be isolated	14 days	From the literature [18]
Time to disease development from initial symptoms	2 days	From the literature [18]
Recovery time	14 days	From the literature [19]
Ratio: Serious symptoms/All symptoms	20%	From the literature [18]
Ratio: Mild symptoms/All symptoms	40%	From the literature [18]
Diagnosis (polymeric chain reaction virus test) efficiency	500 (Tokyo), 110 (Osaka), and 100 (Hokkaido) subjects per day	Peak test numbers collected from MHLW website [20]
Hospital beds	2000 (Tokyo), 1100 (Osaka), and 500 (Hokkaido)	Survey by the Ministry of Health, Labor and Welfare on 1 May 2020 [20]
Infection per day per virus carrier	0.25	Reproduction number 3.3 was divided by average exposure by carrier (5 days incubation and 60% probability for 14 days inapparent infection results in 13.4 days)

**Table 2 ijerph-17-06238-t002:** Epidemic parameters and factors that potentially have a relationship to disease transmission.

	Tokyo	Osaka	Hokkaido
Disease Transmission Parameters			
Baseline relative transmission efficiency	1.00	0.90	0.32
Maximum intervention Effect	0.75	0.83	0.69
Intervention effect before holiday season	0.75	0.40	0.32
Demographics			
Total Population ^*1^	1.39 × 10^7^	8.81 × 10^6^	5.25 × 10^6^
Population density in densely inhabited district ^*2^	1.23 × 10^4^	9.32 × 10^3^	5.09 × 10^3^
Proportion of population in densely inhabited district ^*2^	0.984	0.957	0.752
Behavior-related data			
Maximum Reduction in Outings ^*3^	0.56	0.46	0.34
Maximum Reduction in Outings before holiday season ^*3^	0.55	0.43	0.29
Worker population	7.44 × 10^6^	4.71 × 10^6^	2.66 × 10^6^
Located companies	2.02 × 10^5^	1.05 × 10^5^	6.97 × 10^4^

*^1^ Year 2019. *^2^ Year 2015. *^3^ Expressed as relative to previous year, averaged by week.

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
