# Peer review of "Effectiveness of Social Measures against COVID-19 Outbreaks in Selected Japanese Regions Analyzed by System Dynamic Modeling"

_ijerph, 2020, doi:10.3390/ijerph17176238_

Round 1

Reviewer 1 Report

The manuscript utilizes a great deal of available data to develop models that make accurate predictions of the observed COVID trends in the spring in Japan. Overall it is an interesting model utilizing the available data for this country.

- The introduction contains out-of-date information, mostly citing numbers from April. It is important to make this as current as possible for the best context of the current use of the information, even if the model is based on the earlier data.

- Could the datasets described in 2.1 - 2.4 (PCR results by day per location, population movements data, medical and social components) be provided as supplementary tables? This would allow readers to take advantage of the substantial data compilation efforts that the authors have went to, and so that they can attempt to replicate the model for their own work based on a training dataset. This comprehensive dataset would be a valuable outcome of the paper on its own.

- Some of the arrow lines in Figure 1 do not connect cleanly (especially above “Recovery”). This figure could be redrawn with cleaner lines in MS Powerpoint, if editing the current version is difficult.

- Please format table 2 with no vertical lines, and horizontal lines only separating major headings (“Disease transmission parameters”, “Demographics”, “Behavior-related data. This will make it easier to read. Removing vertical lines and reducing the font size on Table 1 (so that it fits on one page) will likewise improve readability.

Author Response

August 14, 2020

Dear Reviewer,

First of all, we’d like to express to you our gratitude. Your comments and suggestions are invaluable for us to improve our paper.

We believe that we followed all of your comments and suggestions as below, and we hope that our responses meet your expectations and your intentions.

We highly appreciate your cooperation.

Warm Regards,

Kota Kodama, PhD

Graduate School of Technology Management, Ritsumeikan University

2-150, Iwakura-cho, Ibaraki, Osaka, 567-8570, Japan

+81-72-665-2448

[email protected]

ID

Comments and Suggestions

Response

Reviewer

1-1

The introduction contains out-of-date information, mostly citing numbers from April. It is important to make this as current as possible for the best context of the current use of the information, even if the model is based on the earlier data.

Detailed description on health systems was moved to Discussion section to simplify the Introduction. Current information is added in Discussion section.

Line 265-268 in the revised manuscript:

“As of April 2020, there were 12,500 beds for novel infectious diseases nationwide, while there were 10,000 patients in Japan. Although 31,383 hospital beds were ensured by May 21, 2020, health systems are still at risk.”

In addition, the latest information about disease transmission (Reference 30) is added in Discussion section.

Line 289 in the revised manuscript:

“In addition, association of human activity and temperature or humidity is possible.”

Line 299-301 in the revised manuscript:

“The hypothetical mechanism for the associations between population density and transmission proposed by Rubin et al. is that increased droplet transmission and potentially airborne transmission in closer proximity.”

Possibility that local government’s virus testing policy deviated over time was considered in the limitation section.

Line 320-321 in the revised manuscript:

The model was constructed based on the fundamental monitor and control strategy in Japan but detailed approach may have slightly changed over time based on local government’s policy.

Finally, the prospects of working with latest information is added in the end of discussion.

Line 334-336 in the revised manuscript:
“Further analysis using newly collected epidemic data and more detailed social activity data is warranted in the future.”

Reviewer

1-2

Could the datasets described in 2.1 - 2.4 (PCR results by day per location, population movements data, medical and social components) be provided as supplementary tables? This would allow readers to take advantage of the substantial data compilation efforts that the authors have went to, and so that they can attempt to replicate the model for their own work based on a training dataset. This comprehensive dataset would be a valuable outcome of the paper on its own.

PCR results per location was summarized as appendix table 1. As results by day seemed to be lengthy, comprehensive data in every 5 days are provided. This will provide the guide for model reproduction.

Medical and social components already have been shown in Table 1.

Reviewer

1-3

Some of the arrow lines in Figure 1 do not connect cleanly (especially above “Recovery”). This figure could be redrawn with cleaner lines in MS Powerpoint, if editing the current version is difficult.

Figure 1 is re-organized using MS Powerpoint.

Reviewer

1-4

Please format table 2 with no vertical lines, and horizontal lines only separating major headings (“Disease transmission parameters”, “Demographics”, “Behavior-related data. This will make it easier to read. Removing vertical lines and reducing the font size on Table 1 (so that it fits on one page) will likewise improve readability.

Table 2: Vertical lines were removed. Horizontal lines were also removed except lines separating major headings.

Table 1: Vertical lines were removed and font size was reduced.

Reviewer 2 Report

This is a novel and interesting paper. You have been very specific in your detailing of the limitations of the study, suggesting (to me at least) that its publication may well stimulate the further investigations you propose.

I do stand by my opinion that the authors of this paper have undertaken a novel approach to the analysis of multiple data sets and that the paper merits publication for the insights it brings to the challenges of managing COVID-19 spread despite its language idiosyncrasies. With this in mind, I tender the following comments:

  1. Title. It is suggested that the authors reword the title to address grammatical / vocabulary issues. Specifically, to replace 'Japanese several regions' with 'selected Japanese regions'.
  2. Abstract. It is suggested that authors reword the final sentence (lines 29-30) to indicate that their findings may contribute to an understanding of the development of social resilience to future infectious disease threats.

  1. Introduction. The Introduction section contains considerably more words (approx. 1175) than the Discussion section (approx. 670). It is suggested that the authors slightly reorganise the Introduction and consider using the information contained in the text between lines 42 and 62 of the Introduction (commencing with:  "Many countries including China..." and concluding with "...virus testing was considered." in the Discussion section , to enable comparison / contrast of the Japanese experience in hospital bed capacity and utilisation with that of other countries.

  1. Results. It is suggested that the authors revise some of the terminology employed in Figure 6 and throughout the text. Specifically: replacement of 'Enlightenment' with Awareness Raising' in Figure 6 and in lines 146 and 306 ; replacement of  'Sanitary' with 'Hygiene' in Figure 6 and throughout the text (Lines 147, 289 and 306)

  1. Discussion. Please see comment 3 above about reworking the information and references in these Introduction paragraphs into your discussion on the comparative effectiveness of prevention measures as related to hospital bed capacity and utilisation.

  1. Other. In addition to changes to the Abstract suggested in comment 2, perhaps the authors could consider a complete re-phrasing of the Abstract to more clearly indicate the logic of the modelling process. I have taken the liberty of proposing what this could be in the text below. However, this is for the authors at their discretion to determine:

"In Japan's response to coronavirus disease 2019 (COVID-19), virus testing was limited to 17 symptomatic patients due to limited capacity, resulting in uncertainty regarding the spread of infection and the appropriateness of countermeasures. System dynamic modelling, comprised of stock flow and infection modelling, was used to describe regional population dynamics and estimate assumed region-specific transmission rates. The estimated regional transmission rates were then mapped against actual patient data throughout the course of the interventions. This modelling, together with simulation studies, demonstrated the effectiveness of inbound traveller quarantine and resident self-isolation policies and practices. A causal loop approach was taken to link societal factors to infection control measures.  This causal loop modelling suggested that the only effective measure against COVID-19 transmission in the Japanese context was intervention in the early stages of the outbreak by national and regional governments, no social self-strengthening dynamics were demonstrated. These findings may contribute to an understanding of how social resilience to future infectious disease threats can be developed."

I do trust that these will indeed be useful to the authors in the refinement of their text and also for your Editorial review and assessment.

Author Response

2020年8月14日

レビュー担当者様

まず、感謝の意を表したいと思います。あなたのコメントと提案は、私たちが論文を改善するために非常に貴重です。

私たちは、以下のようにすべてのコメントと提案に従っていると信じており、私たちの回答があなたの期待と意図に応えることを願っています。

何卒よろしくお願い申し上げます。

敬具、

児玉光太博士

立命館大学大学院工学研究科

567-8570大阪府茨木市岩倉町2-150

+81-72-665-2448

[email protected]

ID

Comments and Suggestions

Response

Reviewer

2-1

Title. It is suggested that the authors reword the title to address grammatical / vocabulary issues. Specifically, to replace 'Japanese several regions' with 'selected Japanese regions'.

We have reworded the title as suggested.

The revised title is as follows:

“Effectiveness of social measures against COVID-19 outbreaks in selected Japanese regions analyzed by system dynamic modeling”

Reviewer

2-2

Abstract. It is suggested that authors reword the final sentence (lines 29-30) to indicate that their findings may contribute to an understanding of the development of social resilience to future infectious disease threats.

We re-phrased the abstract taking the proposed example (reviewer comment 2-6) into account. The final sentence should be as follows:

These findings may contribute to an understanding of how social resilience to future infectious disease threats can be developed.

Reviewer

2-3

Introduction. The Introduction section contains considerably more words (approx. 1175) than the Discussion section (approx. 670). It is suggested that the authors slightly reorganise the Introduction and consider using the information contained in the text between lines 42 and 62 of the Introduction (commencing with:  "Many countries including China..." and concluding with "...virus testing was considered." in the Discussion section , to enable comparison / contrast of the Japanese experience in hospital bed capacity and utilisation with that of other countries.

We have re-organized the introduction and discussion as suggested.

1) Texts existed between lines 42 and 62 in original manuscript were moved to the Discussion section.

2) Introduction was re-organized. The sentences that comes after removed paragraph is Re-constructed as follows:

“In handling the complex COVID-19 transmission processes in the population and the effects of societal factors, the idea to use system dynamics, describing complex social systems as a collective set of mathematical equations, was drawn based on some considerations.”

Reviewer

2-4

Results. It is suggested that the authors revise some of the terminology employed in Figure 6 and throughout the text. Specifically: replacement of 'Enlightenment' with Awareness Raising' in Figure 6 and in lines 146 and 306; replacement of 'Sanitary' with 'Hygiene' in Figure 6 and throughout the text (Lines 147, 289 and 306)

All “Enlightenment” were replaced with “Awareness rising”. They appear in lines 123 and 305, and in Figure 6 in revised manuscript.

All “Sanitary” were replaced with “Hygiene”. They appear in lines 124, 287 and 305, and in Figure 6.

Reviewer

2-5

Discussion. Please see comment 3 above about reworking the information and references in these Introduction paragraphs into your discussion on the comparative effectiveness of prevention measures as related to hospital bed capacity and utilisation.

We have re-organized the introduction and discussion as suggested.

1) Texts existed between lines 42 and 62 in original manuscript were moved to the Discussion section.

2) Discussion was re-organized using the information originally existed between lines 42 and 62 in original manuscript.

Reviewer

2-6

Other. In addition to changes to the Abstract suggested in comment 2, perhaps the authors could consider a complete re-phrasing of the Abstract to more clearly indicate the logic of the modelling process. I have taken the liberty of proposing what this could be in the text below. However, this is for the authors at their discretion to determine:

"In Japan's response to coronavirus disease 2019 (COVID-19), virus testing was limited to symptomatic patients due to limited capacity, resulting in uncertainty regarding the spread of infection and the appropriateness of countermeasures. System dynamic modelling, comprised of stock flow and infection modelling, was used to describe regional population dynamics and estimate assumed region-specific transmission rates. The estimated regional transmission rates were then mapped against actual patient data throughout the course of the interventions. This modelling, together with simulation studies, demonstrated the effectiveness of inbound traveller quarantine and resident self-isolation policies and practices. A causal loop approach was taken to link societal factors to infection control measures.  This causal loop modelling suggested that the only effective measure against COVID-19 transmission in the Japanese context was intervention in the early stages of the outbreak by national and regional governments, no social self-strengthening dynamics were demonstrated. These findings may contribute to an understanding of how social resilience to future infectious disease threats can be developed."

We re-phrased the abstract taking the proposed example into account as follows:

In Japan's response to coronavirus disease 2019 (COVID-19), virus testing was limited to symptomatic patients due to limited capacity, resulting in uncertainty regarding the spread of infection and the appropriateness of countermeasures. System dynamic modelling, comprised of stock flow and infection modelling, was used to describe regional population dynamics and estimate assumed region-specific transmission rates. The estimated regional transmission rates were then mapped against actual patient data throughout the course of the interventions. This modelling, together with simulation studies, demonstrated the effectiveness of inbound traveler quarantine and resident self-isolation policies and practices. A causal loop approach was taken to link societal factors to infection control measures. This causal loop modelling suggested that the only effective measure against COVID-19 transmission in the Japanese context was intervention in the early stages of the outbreak by national and regional governments, and no social self-strengthening dynamics were demonstrated. These findings may contribute to an understanding of how social resilience to future infectious disease threats can be developed.